# HIV-1 subtype A1, D, and recombinant proviral genome landscapes during long-term suppressive therapy

Guinevere Q. Lee [1] ✉, Pragya Khadka[1], Sarah N. Gowanlock[2], Dennis C. Copertino Jr [1], Maggie C. Duncan[3,4], F. Harrison Omondi [3,4], Natalie N. Kinloch[3,4], Jingo Kasule[5], Taddeo Kityamuweesi[5], Paul Buule[5], Samiri Jamiru[5], Stephen Tomusange[5], Aggrey Anok[5], Zhengming Chen[6], R. Brad Jones [1], Ronald M. Galiwango[5], Steven J. Reynolds[5,7,8], Thomas C. Quinn[7,8], Zabrina L. Brumme [3,4], Andrew D. Redd[7,8,9,11] & Jessica L. Prodger[2,10,11]

The primary obstacle to curing HIV-1 is a reservoir of CD4+ cells that contain stably integrated provirus. Previous studies characterizing the proviral landscape, which have been predominantly conducted in males in the United States and Europe living with HIV-1 subtype B, have revealed that most proviruses that persist during antiretroviral therapy (ART) are defective. In contrast, less is known about proviral landscapes in females with non-B subtypes, which represents the largest group of individuals living with HIV-1. Here, we analyze genomic DNA from resting CD4+ T-cells from 16 female and seven male Ugandans with HIV-1 receiving suppressive ART ($n = 23$). We perform near-full-length proviral sequencing at limiting dilution to examine the proviral genetic landscape, yielding 607 HIV-1 subtype A1, D, and recombinant proviral sequences (mean 26/person). We observe that intact genomes are relatively rare and clonal expansion occurs in both intact and defective genomes. Our modification of the primers and probes of the Intact Proviral DNA Assay (IPDA), developed for subtype B, rescues intact provirus detection in Ugandan samples for which the original IPDA fails. This work will facilitate research on HIV-1 persistence and cure strategies in Africa, where the burden of HIV-1 is heaviest.

The HIV-1 latent viral reservoir (LVR) is primarily made up of resting CD4+ (rCD4) T-cells containing stably integrated HIV-1 provirus, and is the main barrier to a cure for HIV-1[1]. The vast majority (>90%) of stably infected cells contain defective proviruses harboring large deletions, premature stop codons, or hypermutations[2,3]. Elimination or deactivation of genome-intact proviruses is the primary goal of HIV-1 cure research. To date, the vast majority of research examining HIV-1 proviral genome-intactness has been done using samples from males with

[1]Department of Medicine, Division of Infectious Diseases, Weill Cornell Medicine, New York, NY, USA. [2]Department of Microbiology and Immunology, Western University, London, ON, Canada. [3]Faculty of Health Sciences, Simon Fraser University, Burnaby, BC, Canada. [4]British Columbia Centre for Excellence in HIV/AIDS, Vancouver, BC, Canada. [5]Rakai Health Sciences Program, Kalisizo, Uganda. [6]Department of Population Health Sciences, Division of Biostatistics, Weill Cornell Medicine, New York, NY, USA. [7]Division of Intramural Research, National Institute of Allergy and Infectious Diseases, National Institutes of Health, Bethesda, MD, USA. [8]Department of Medicine, Johns Hopkins University School of Medicine, Baltimore, MD, USA. [9]Institute of Infectious Disease and Molecular Medicine, University of Cape Town, Cape Town, South Africa. [10]Department of Epidemiology and Biostatistics, Western University, London, ON, Canada. [11]These authors contributed equally: Andrew D. Redd, Jessica L. Prodger. ✉e-mail: gul4001@med.cornell.edu

HIV-1 subtype B, the primary subtype found in the United States and Europe, even though globally, the majority of people with HIV-1 (PWH) have non-B subtypes[2,4,5].

HIV-1 subtypes A1 and D predominate in Uganda and other countries in East Africa, where females are also heavily impacted. Previous work has shown possible biological differences between HIV-1 subtypes, with subtype D being associated with faster disease progression, faster rate of CD4+ T-cell decline, lower pre-treatment CD4+ T-cell count, and higher mortality[6–10]. Subtype D HIV-1 has also been associated with greater replicative fitness, increased apoptosis of CD4+ T-cells, and higher frequencies of CXCR4 co-receptor usage relative to subtypes A1 and/or B[11–13]. Given HIV-1 subtype differences in pathogenesis and genetics, viral subtypes may have different proportions of defective proviruses and patterns of genomic areas with deletions or hypermutation. Indeed, previous studies have suggested potential differences in reservoir sizes across viral subtypes[14,15]. An in-depth examination of the composition of reservoirs across viral subtypes is, therefore, critical to understand the biology of HIV-1 persistence and assess if subtype-specific strategies to cure HIV-1 are required. It is also essential to inform the design of assays to quantify genome-intact proviruses—such as the Intact Proviral DNA Assay (IPDA)—in diverse populations. The IPDA is a dual-target droplet digital PCR (ddPCR) assay that features parallel independent reactions to quantify HIV-1 and human (RPP30 gene) DNA[16]. Through the detection of two regions in the HIV-1 genome, most defective provirus sequences can be identified, allowing the quantification of the number of intact and total (i.e., both intact and defective) HIV-1 genomes per million CD4+ T-cells[16]. However, the IPDA was originally designed using subtype B sequences. Establishing the suitability of the primer/probe locations and sequences for use in Africa, which is affected almost exclusively by non-B subtypes, will accelerate cure research in the regions where the HIV-1 disease burden is the heaviest globally. To date, studies have adapted the IPDA for cross-subtype applications inclusive of subtypes A, B, C, D, and CRF01_AE, and for subtypes B and C[17,18]. Given that HIV-1 genetic diversity is often region-specific, we characterized viral DNA genome sequences in people living with HIV in Rakai, Uganda, who were receiving suppressive ART, and leveraged these to design a regionally adapted IPDA that would maximally capture local HIV sequence variation.

## Results

### Study population
Of the 90 PWH in the Rakai Health Sciences Program (RHSP) LVR cohort, 23 individuals were selected for this study to represent the cohort's sex and HIV-1 subtype distribution. HIV-1 subtype was initially assessed using short-amplicon sequencing of reverse transcriptase (RT; HXB2 bases = 2723–3225) and gp41 (HXB2 bases = 7938–8256) from viral outgrowth assays[19]. Based on this initial subtyping, the subtype distribution of participants included in this study was A1 ($n = 6$), C ($n = 1$), D ($n = 11$) and A1/D recombinant ($n = 5$). Median age was 42 (range 29–53, IQR 38–45) years, and 16/23 participants (70%) were female. Participants had been receiving suppressive antiretroviral therapy (ART) for a median of nine years (IQR 6–11) and had a median CD4+ T-cell count of 761 cells/μL (IQR 586–892) (Supplementary Table S1). Replication-competent HIV-1 reservoir sizes at sampling were a median of 0.5 (IQR 0.3–1.3) infectious units per million cells (IUPM) as measured by the Quantitative Viral Outgrowth Assay (QVOA). The median total HIV-1 DNA load was 1316 copies per million cells (IQR 786–2016) as measured by ddPCR (Supplementary Table S2).

### FLIP-seq near-full-length HIV-1 DNA genome sequencing improved viral subtyping resolution
Using full-length individual proviral sequencing (FLIP-seq), 607 near-full-length single-genome-amplified (SGA) HIV-1 DNA genome sequences were obtained after sampling a median of 120,000 cells

(IQR 71,000–190,000 cells) per donor (median 22 genomes per donor, IQR 19–28; Supplementary Table S2). Each FLIP-seq-derived genome was subjected to viral subtyping (Supplementary Fig. S1 and Table S2) and compared to donor-matched QVOA-derived short-amplicon RT/gp41 sequences. FLIP-seq confirmed the original five A1/D recombinants identified by the short-amplicon RT/gp41 approach as inter-subtype recombinants, and identified five additional inter-subtype-recombinant infections that the original short-amplicon approach had identified as pure subtypes (Fig. 1 and Supplementary Fig. S1). Among the ten individuals with inter-subtype recombinant HIV-1 by FLIP-seq ($n = 9$ A1/D, $n = 1$ A1/C/D), the approximate recombination regions, defined by two adjacent genes mapping to different subtypes, were distinct across individuals (between-host). Moreover, seven of these ten individuals had clear evidence that recombination regions were conserved within-host, consistent with the acquisition of a recombinant infection.

### Intact proviral genomes were rare in subtype A1, D, and recombinant infections
The 607 HIV-1 DNA genomes were examined for intactness using HIVSeqinR (Fig. 2 shows two representative donors) and the proportions of intact genomes within the total HIV-1 DNA pool was calculated (Fig. 3A, B). Intact genomes were detected in eight individuals (32/607, 5% of the total HIV genomes obtained). In these eight participants, the proportion of intact HIV-1 genomes was highly variable, ranging from <2 to 33% between donors (Supplementary Tables S2, S3). Many defective proviruses detected (495/575, 86%) contained large deletions (range 33–100% between donors). The second most common genome defect observed was APOBEC-3G/3F-associated hypermutations (50/575, 9%; 2–58% between donors), followed by packaging signal 5' defects (18/575, 3%; 0–14% between donors), internal inversions (6/575, 1%; 0–7% between donors), and premature stop codon(s) in *gag/pol/env* (4/575, 1%; 0–5% between donors; Fig. 3B and Supplementary Table S3). When grouped by subtype, the relative distribution of intact and defective genome categories was not significantly different between subtype A1 (4 donors, 94 sequences, 6.4% intact), subtype D (9 donors, 234 sequences, 8.1% intact) and A1/D recombinants (9 donors, 249 sequences, 2.4% intact; Fisher's exact test $p = 0.3$; Table S4). Genomic deletions were most frequently observed in the region between RT and *env* (Supplementary Fig. S2).

### Clonal expansion of infected cells was detected across subtypes A1, D, and recombinant HIV-1
We have previously validated that when two HIV-1 DNA genomes share 100% sequence identity by FLIP-seq, they also share identical integration sites, which was used in this study as a marker of inferred clonal expansion of infected cells[20]. Despite the relatively shallow sampling depth per donor, inferred clonal expansion was detected in 15/23 donors (Fig. 3C and Supplementary Table S5). Inferred defective clones were most commonly detected and made up <3 to 70% of the intra-host proviral DNA population, whereas inferred intact clones were detected in only two donors at 13 and 14% of the population. In a pooled analysis, subtype A1 had significantly lower overall frequency of inferred clonal expansion (5% of all sequences; 4 donors, 94 sequences) relative to subtype D (26%; 9 donors, 234 sequences; odds ratio 0.16; Fisher's exact $p < 0.0001$) and A1/D recombinants (16%; 9 donors, 249 sequences; odds ratio 0.29; Fisher's exact $p = 0.007$), but these observations may be biased by the lower number of study donors with subtype A1 infection (Supplementary Table S6).

### IPDA target assessment: genomic location
To explore appropriate IPDA amplicon positioning for discrimination of intact proviruses in the RHSP cohort, we performed a sliding window analysis similar to that performed during the development of the

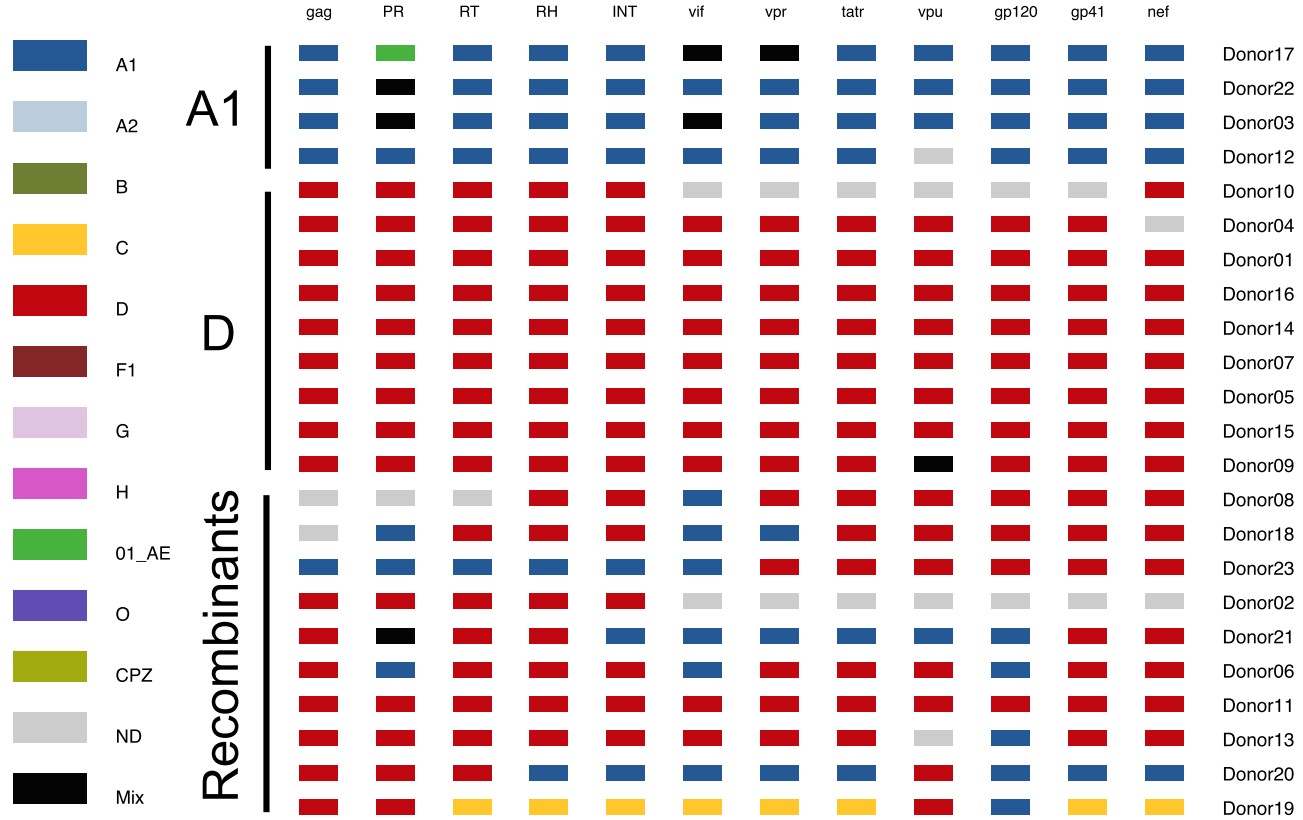

**Fig. 1 | HIV subtype distribution among the 23 donors from Rakai Uganda.** Only the longest HIV genome recovered per donor is plotted here. See Fig. S1 for a comprehensive view and subtype interpretations for each of the 607 viral genomes examined in this study. As described in Fig. S1, in Donor 17's representative sequence (first row) has a protease (PR) region that best matches the subtype 01_AE consensus (green), but this was interpreted as subtype A1 because this genomic region of 01_AE maps to pure A1 by the Los Alamos HIV Sequence Database Recombinant Identification Program (RIP). PR protease, RT reverse transcriptase, RH RNAseH, INT integrase, tatr tat rev exon1.

original assay, here called IPDA-B[16]. This analysis revealed that the original IPDA-B amplicon regions (HXB2 coordinates *psi* 692–797 and *env* 7736–7851) should correctly identify 97.3% of proviruses with large deletions in the RHSP cohort (Fig. 4A). Although other regions had slightly higher predictive values for proviruses with large deletions (for example, HXB2 1778–2048 and 8288–8408 should exclude 98.8% of proviruses with large deletions), the original IPDA-B *psi* amplicon location has the additional benefit of containing the major splice donor site (MSD; HXB2 743), a common location of small deletions that render proviruses defective[4,21,22]. Importantly, 100% (32/32) of the intact genomes contained the original IPDA-B *psi* and *env* binding sites, yielding an assay sensitivity estimate of 100% when considering primer/probe location only. Moreover, only 3% (19/575) of defective genomes were observed to contain both of the IPDA-B binding sites but were not hypermutated, yielding an assay specificity estimate of 97% when considering primer/probe location only. We also observed two genomes (subtype A1/D and A1/C/D) with internal repeats resulting in duplicated *psi* binding sites. This could increase fluorescence amplitudes in the IPDA, but should not influence intact reservoir quantification. Primer/probe location-based sensitivity and specificity estimates were comparable across subtypes (Supplementary Fig. S3 and Table S7). Given the results of these analyses, we retained the same primer and probe locations as the original IPDA-B.

### Adaptation of the IPDA for use in the RHSP cohort

We next adapted the IPDA-B primers/probes to the HIV sequence diversity observed in the RHSP cohort. Indeed, all HIV-1 subtypes represented in the present study had at least one mismatch against the published IPDA-B primers/probes (Fig. 4B). Moreover, at base 13 of the

IPDA-B *env* probe (HXB2 7781–7798, which features an Adenine (A) in the competitive probe that discriminates hypermutated proviruses in subtype B), intact subtype A1 sequences naturally harbor an A rather than the consensus Guanine (G) in subtype B (Supplementary Fig. S4). As expected, the IPDA-B was able to quantify intact and total proviruses in HIV-1 subtype B samples, but it failed when applied to at least four non-B samples from the RHSP cohort (examples in Fig. 4C). Examination of these sequences revealed polymorphisms that likely caused the IPDA-B failure (Fig. 4D). We therefore modified the original IPDA-B primers and probes by introducing degenerate bases at common polymorphic sites to account for the subtype A1/D diversity observed in the $n = 577$ subtype A1, D, and recombinant sequences collected in this study (sequences from the single participant with recombinant A1CD HIV-1 were excluded), while maintaining ddPCR assay specifications. We refer to these adapted primers/probes, which are listed in Supplementary Table S8, as the IPDA-A1D. We retained the original IPDA-B unlabeled hypermutation probe design because, similar to subtype B sequences, 96% of hypermutated sequences in the RHSP cohort had G-to-A mutations at both positions 5 and 13 of the *env* probe, which are the two hypermutation-discrimination sites in the IPDA-B assay. As mentioned above, intact subtype A1 sequences also naturally harbor an A at position 13 of the *env* probe, so we also needed to confirm that IPDA-A1D could discriminate hypermutated from non-hypermutated subtype A1 sequences based on a single G-to-A mutation at the *env* hypermutation probe position 5.

### IPDA-A1D validation

We first confirmed that the degenerate bases within the *psi* and *env* probes would not affect signal amplitude to the point of

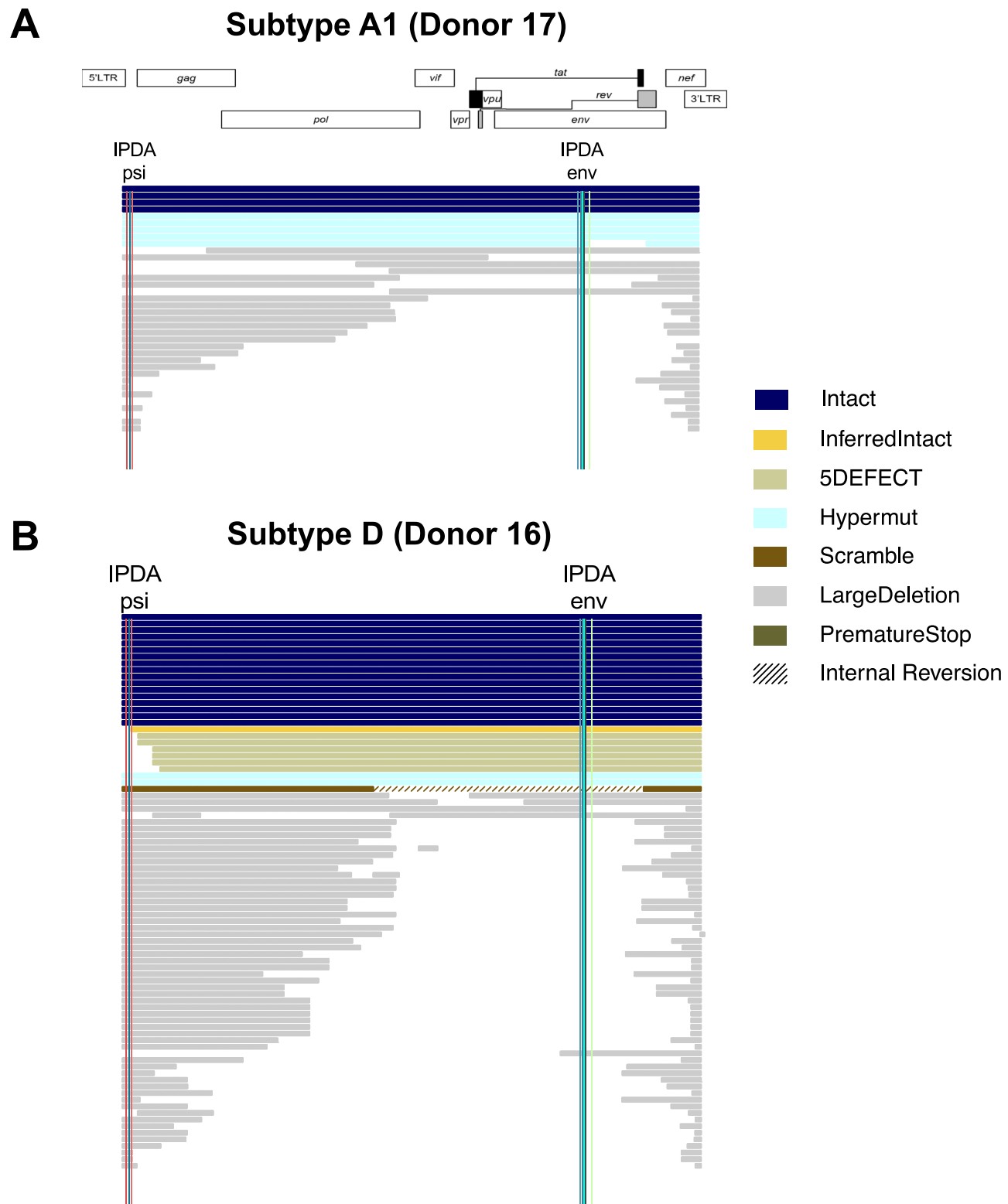

**Fig. 2 | HIV-1 DNA near-full-genome (HXB2 638–9632) landscape of two representative donors. A** Donor 17, HIV-1 subtype A1; **B** Donor 16, subtype D. Each horizontal line represents a viral genome captured via FLIP-seq and single-genome amplification (SGA). The genome-intactness inference was performed using HIV-SeqinR. Briefly, "LargeDeletion" is defined as any genome <8000 base pairs, "Internal Reversion" is any genome with internal reversions, "Scramble" is any genome with regions that are out-of-order (e.g., part of *gag* located downstream of *pol*), "Hypermut" is any genome with APOBEC-3G/3 F-associated hypermutations,

"PrematureStop" is any genome with a premature stop codon in *gag, pol,* or *env*, "5DEFECT" is any genome with ≥15 nucleotide insertion/deletions relative to HXB2 and NL4-3 between genomic coordinates 638–789. Any genomes without these defects are classified as "intact." "Inferred Intact" refers to genomes that are missing 5′ sequence information (likely due to sequencing artifacts) but are otherwise intact. Vertical lines mark the approximate locations of IPDA *psi* (red and blue) and *env* (green and teal) primer and probe locations.

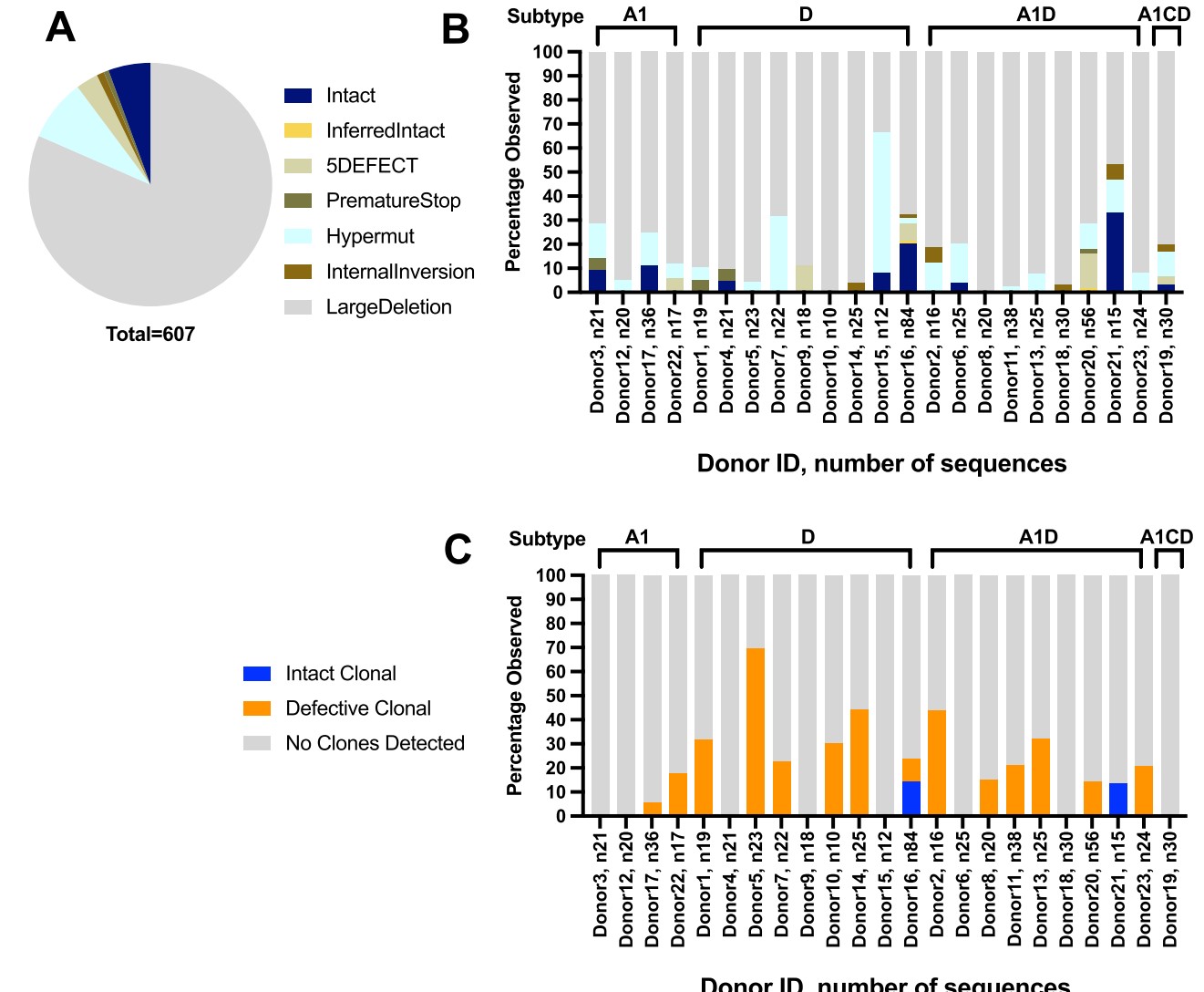

**Fig. 3 | Proviral defects and clonality. A** Intact genomes were rare (6%) among the 607 HIV DNA genomes obtained from individuals living with HIV on long-term ART in Rakai, Uganda. The most common defects were large deletions (86%). **B** Each study donor, regardless of viral subtype, had a distinct profile of intact (navy blue) versus defective genome categories. **C** Clonal expansion of infected cells was detected in both intact (blue) and defective (orange) genomes.

compromising quantification. To accomplish this, we quantified intact proviral copies in J-Lat cells[23] (which contain provirus matching the IPDA-B sequences), using the original IPDA-B primers/probes, as well as the IPDA-B primers combined with the degenerate IPDA-A1D probes. While the IPDA-A1D probes did reduce the fluorescence amplitude in both the *psi* and *env* channels, this reduction did not compromise our ability to discriminate positive from negative droplets, nor did it affect our ability to quantify intact proviruses (Supplementary Fig. S5A). Next, to test whether IPDA-B and IPDA-A1D yield comparable reservoir size measurements, we applied the IPDA-B and IPDA-A1D assays to a panel ($n = 13$) of cell lines and clinical subtype B HIV-1 samples whose sequences were predicted to be detectable by both assays. We observed an excellent correlation between intact proviral measurements (Spearman correlation, $\rho = 0.96$; $p < 0.0001$; Supplementary Fig. S5B), with no significant bias towards higher (or lower) measurements in either assay (Wilcoxon signed-rank test, $p = 0.07$; Supplementary Fig. S5C). To test whether IPDA-A1D would cross-react with human DNA, we tested the assay in $n = 5$ HIV-seronegative individuals, none of which yielded a signal (representative plot in Supplementary Fig. S6). To test whether the unlabeled hypermutation probe could correctly screen out

hypermutated subtype A1 and D genomes, we applied IPDA-A1D to purified amplicons representing near-full-length hypermutated viral genomes, one from each of subtypes A1 and D. We verified that the assay correctly returned only droplets positive in the *psi* region, and not in the *env* region (Supplementary Fig. S7A, B). To test whether the unlabeled hypermutation probe would compete with the labeled probe for binding to intact genomes and thereby reduce signal, a particular concern for subtype A1, we applied the IPDA-A1D assay to two purified amplicons representing near-full-length intact subtype A1 and D genomes, with and without the hypermutation probe included in the reaction (Supplementary Fig. S7C). While the inclusion of the hypermutation probe modestly reduced signal amplitude in the *env* channel, this reduction did not compromise our ability to discriminate positive and negative droplets, and the number of intact HIV-1 copies was essentially identical regardless of the inclusion of the hypermutation probe. After completing these validation experiments, we applied the IPDA-A1D to samples from participants in the RHSP cohort for which the IPDA-B failed ($n = 4$). In all cases, the IPDA-A1D rescued the detection of intact proviruses (two representative plots are shown in Fig. 4C, where a detailed analysis of Donor 20's result is provided in Fig. S8). Of these four participants, IPDA-A1D

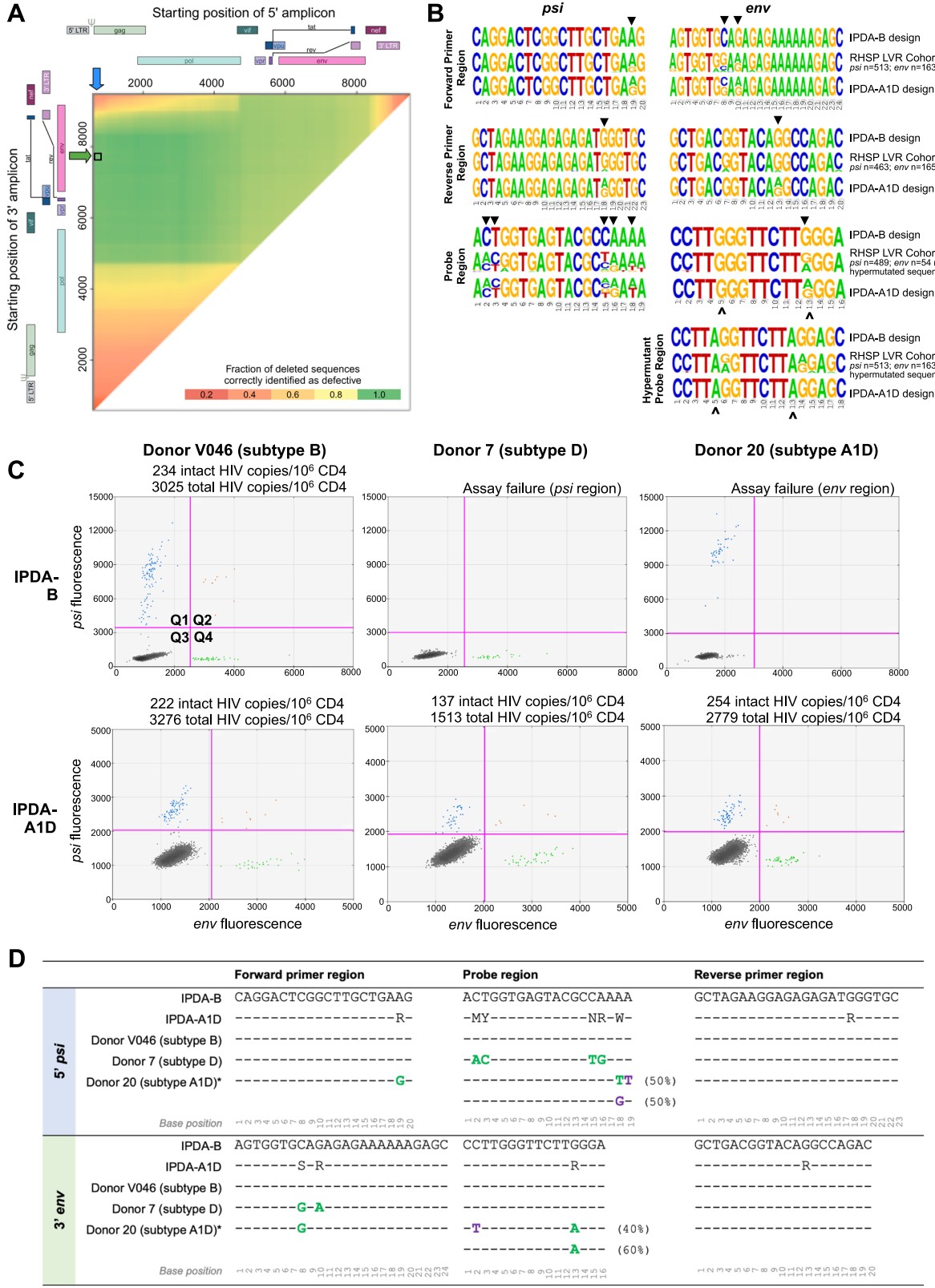

yielded between 97 and 254 intact, and between 1477 and 2779 total, HIV copies per million CD4+ T-cells (Supplementary Table S9).

## Discussion

As HIV-1 cure research progresses, there is an urgent need to understand viral reservoirs formed by strains circulating in Africa, where the disease burden is heaviest globally. Results from this study show that, like subtype B, HIV-1 DNA intact genomes are rare and clonally-expanded genomes are relatively common in people with HIV-1 subtypes A1, D, and recombinants receiving long-term ART[2,4,5,24]. To our knowledge, this is the first study of subtype A1 and D near-full-genome proviral genetics and our sample size of 607 sequences rivals subtype B studies[2,4,5].

**Fig. 4 | IPDA adaptation to subtypes A1/D variants observed in the RHSP cohort. A** Sliding window analysis of 495 proviral genomes with large deletions to identify optimal IPDA amplicon placement for discrimination of sequences with deletions. The black box marks the target regions for both IPDA-B and IPDA-A1D. **B** Diversity within the IPDA *psi* and *env* target regions in the 577 subtype A1/D RHSP sequences (middle row in each set), along with the IPDA-B (top row) and IPDA-A1D (bottom row) oligo designs. Arrowheads (▼) indicate degenerate base sites in the IPDA-A1D oligos. Accent marks (^) identify hypermutation-discrimination positions within *env* and hypermutant probes. **C** ddPCR plots showing *psi*-single-positive (Q1, blue), *psi*- and *env*- double-positive (Q2, orange), double-negative (Q3, gray), and *env* single-positive (Q4, green) events for three participants, along with the IPDA assay results. Plots show a single representative replicate. **D** *Psi* (5′ target) and *env* (3′ target) region sequences for the IPDA-B and IPDA-A1D, and HIV sequences that are intact in these regions from the donors shown in (C). Dashes (−) indicate a match to the IPDA-B reference sequence. Green bases identify donor mismatches to the IPDA-B that are captured by IPDA-A1D. Donor polymorphisms not captured by IPDA-A1D are in purple. Note that all oligo regions in this figure are depicted in the forward direction; see Table S8 for the actual sequences. * Note that donor 20 had within-host HIV genetic variation in both the *psi* and *env* probe target regions. The table summarizes the observed sequences, where the number in brackets indicates the frequency whereby each was observed.

Our observation that subtype A1, D, and recombinant proviral sequences are rarely genome-intact suggests that HIV-1 persistence research in Africa will face the same needle-in-a-haystack challenge in identifying cells infected with genome-intact HIV-1 as in subtype B. Our findings also suggest that inferred clonal expansion of infected cells is likely a major mechanism of persistence in non-subtype B HIV-1.

Near-full-length HIV-1 genome sequencing identified a larger number of inter-subtype recombinant genomes when compared with subtyping based on short-range sequencing of RT and gp41. The high proportion of people with intra-subtype recombinant HIV-1 is consistent with our previous observations[25] and with data from previous Ugandan HIV-1 sequencing studies from the PANGEA consortium and the UARTO cohort[26–29]. Our results also further underscore the subtype complexity in this geographical region, as evidenced by the lack of any officially named circulating recombinant forms, and the lack of common inter-subtype recombination locations among the ten individuals with recombinant HIV-1. Any reservoir measurement assay that relies on molecular methods to detect HIV-1 will need to take this diversity into account. Furthermore, these observations suggest that subtype-specific assays may have limited utility in areas where multiple subtypes and recombinants co-circulate.

Furthermore, our finding that each recombinant A1/D genome was unique is suggestive of a high frequency of subtype A1 and D dual infection in the region. However, since we intentionally selected participants based on their initial subtyping results, and purposely included individuals with recombinant infections, the subtype distribution in this study is not necessarily geographically representative, and therefore, we cannot use our observations to estimate the level of dual infection in the region. Our findings are nevertheless consistent with our previous observations from an independent cohort, where inter-subtype recombinants with unique breakpoints (as determined by the Los Alamos Recombinant Identification Program, RIP) made up 46% of HIV-1 infections in Mbarara, Uganda between 2005–2010[25]. Future population-level and longitudinal studies will be required to address the prevalence and viral evolution patterns of multi-subtype infections leading to recombination.

We also modified the original IPDA (IPDA-B) to capture the genetic diversity in subtypes A1 and D, which we refer to as the IPDA-A1D. Other IPDA-like assays have been designed for non-subtype B HIV-1, including the cross-subtype IPDA (CS-IPDA)[17], but these designs relied on publicly available and historic sequences from the Los Alamos HIV sequence database, and thus may not reflect the contemporary genetic diversity of circulating variants in this cohort. The 607 FLIP-seq viral DNA genomes in this study enabled us to develop a targeted assay to capture the cohort's viral genetic diversity. Whether IPDA-A1D will be appropriate for subtype A1 and D HIV-1 outside of this cohort should be investigated in future studies.

Our findings have several limitations, many of which are associated with the technical limitations and limited sampling depth inherent in a FLIP-seq-based approach. FLIP-seq uses a column-based kit for genomic DNA extraction, which is prone to genomic DNA shearing, and therefore, there is a possibility that not all full-length viral genomes in the samples were successfully captured[30]. FLIP-seq is also known to amplify genomes with large internal deletions more efficiently than full-length ones[30,31]. Due to these limitations, we have chosen to not use the number of intact genomes per sample to quantify reservoir sizes, but instead focused on characterizing viral DNA genetic diversity in our cohort. In addition, this study examined only 23 individuals with a mean of 26 HIV-1 genomes/person, and only detected 32 intact genomes. Our results are also limited to subtypes A1 and D circulating in the Rakai region of Uganda, and may not be reflective of other regions or other subtypes, importantly HIV-1 subtype C. The use of degenerate probes in the IPDA-A1D to accommodate HIV diversity reduced the separation between negative and positive ddPCR droplets in both *psi* and *env* channels. It is possible that separation will be even further reduced in samples with additional mismatches, an issue that will need to be monitored as we apply IPDA-A1D to larger populations. IPDA results should always be interpreted in context of HIV's substantial between- and within- host diversity, where the latter can lead to quantification of only a portion of a given individual's reservoir, a scenario that cannot be assessed without individual-level HIV sequencing of every participant. As a result, while IPDA is a robust assay for samples where the viral diversity is constant or known, and fairly robust for longitudinal characterization of samples from individuals on long-term ART, caution should be exercised when applying the IPDA cross-sectionally across individuals or cohorts whose HIV sequences have not been characterized. Wherever possible, alternative methods such as viral outgrowth assays, FLIP-seq and/or alternative ddPCR methods that targets different genome regions should be used to support IPDA findings in these cases.

In conclusion, our findings suggest that the composition of the HIV-1 proviral landscape is broadly comparable between subtypes A1, D, and B. The design of approaches to target intact HIV reservoirs in Africa will, therefore, likely face similar challenges to those seen in North America and Europe. Our viral DNA genome sequence data has enabled us to adapt the IPDA to the RHSP cohort, providing an important additional tool to study viral persistence in this region. Future studies should invest in gaining a deeper understanding of reservoirs in non-B subtypes to further evaluate whether there are other subtype-specific differences in viral reservoirs that will play roles in persistence, reactivation, or clearance.

## Methods

### Ethics statement

The study was approved by the National Institute of Allergy and Infectious Diseases (National Institutes of Health), Uganda Virus Research Institute, Uganda National Council for Science and Technology, and Weill Cornell Medicine Institutional Ethics Review Board (20-01021318). Reservoir quantification in subtype B samples was approved by the University of British Columbia/Providence Health Care and Simon Fraser University Research ethics boards. All participants provided written informed consent. Participants were provided an honorarium of 20,000 Ugandan Shillings (~5 US dollars) per completed study visit. In addition, participants were reimbursed between 8000 and 50,000 Uganda shillings (~2–12.5 US dollars) per visit for transportation costs depending on distance and economic conditions.

Total reimbursement may be up to 70,000 Uganda shillings (-17.5 US dollars) per visit.

## Study population and sample collection

Details of the study population were previously presented[14,19]. Briefly, resting CD4+ T-cells (rCD4; CD69-/CD25-/HLA-DR-) were collected during cell purification for the quantitative viral outgrowth assay (QVOA) from participants of the Rakai Health Sciences Program (RHSP) LVR clinical study who had been virally suppressed for >1 year (n = 23)[14]. The participants examined here were initially selected based on (i) the availability of frozen rCD4 cells and (ii) viral subtype data, to obtain a representation of subtype A1, D, and A1/D recombinants. This initial viral subtyping for sample selection was performed by Illumina deep-sequencing of *pol* (HXB2 2723–3225) and gp41 (HXB2 7938–8256) sequences from p24-positive QVOA wells that contained a single dominant viral sequence in both regions[32].

## Near-full-genome HIV-1 DNA sequencing

Total genomic DNA (gDNA) was extracted from rCD4 cells using the QIAGEN AllPrep Mini extraction kit (catalog # 80204), followed by near-full-length single-genome-amplified (SGA) provirus sequencing (FLIP-seq) with a target of ~20 genomes per individual[3,5,30]. Total HIV-1 DNA was quantified in gDNA extracts by BIO-RAD droplet digital PCR (ddPCR) targeting HXB2 684–810, and then gDNA was diluted to one HIV-1 genome per well before near-full-genome nested PCR amplification (HXB2 638–9632) using published primers[5]. Primers were validated against subtype A1 and D HIV-1 (see "ddPCR and FLIP-seq primer/probe validation" section in the supplement, which includes Table S10 and Figs. S9, S10). SGA PCR amplicons were sequenced at the Massachusetts General Hospital Center for Computational and Integrative Biology (CCIB) DNA core. Briefly, partial Illumina-compatible adapters with unique barcodes were ligated onto each sample, after which the full-length adapter sequences were added during a low-cycle PCR amplification step. Libraries were pooled in equimolar concentrations for multiplexed sequencing on the Illumina MiSeq platform with 2 × 150 run parameters. The resulting short reads (average 117,000 per amplicon) were de novo assembled using the DNA core de novo assembler UltraCycler v1.0 (proprietary). The average coverage per position was 2520 reads. At each nucleotide position, the majority base was taken as the final identity of the position. To monitor PCR and/or sequencing errors, the frequency of mismatches at each nucleotide position was calculated: over 99.6% of base positions contained fewer than 5% mismatches to the final assigned base. Though all FLIP-seq viral genome sequences were derived from a limiting dilution step to ensure an 85.7% probability of sequencing one viral DNA template per reaction, it is possible that more than one HIV-1 DNA template was sampled in a minority of reactions[30]. To address this, all FLIP-seq data were subjected to bioinformatics removal of wells suspected to contain >1 HIV-1 genomes[30]. Briefly, a genome would be removed if the deep sequencing results contained non-random base pair mixtures (exceeding 20% read-level mismatches in over 10% of base positions in the genome), and/or assembled into multiple contigs. Our removal rate in this study was 2.7% and was consistent with Poisson statistics. All 607 genomes obtained in this study are available publicly in Gen-Bank (accession numbers OQ686003–OQ686609).

## Bioinformatics determination of viral DNA genome-intactness

Genome-intactness was inferred using a bioinformatics pipeline HIVSeqinR[3] version 2.7.1, after first evaluating the pipeline's performance in classifying subtype A1 and D viral genomes (see "Validation of HIVSeqinR for use in HIV subtypes A1 and D" section of the supplement, which includes Figs. S11–S16). Each genome was also manually inspected for inference accuracy. The full definition of genome-intactness can be found in our previous publications[3,30]. Briefly, we first mapped each viral genome against HIV-1 subtype A1, B, and D reference genomes. Then, we classified any genomes <8000 nucleotides as having "Large Deletions". The remaining genomes, all ≥8000 nucleotides, were checked for the presence of internal inversions and/or gene-region scrambles. Genomes lacking these defects were aligned and checked for the presence of APOBEC-3G/F-associated hypermutations using an in-house adapted Hypermut 2.0 algorithm[33], then each open reading frame was translated. These genomes were classified in the order of "Hypermutated" if they were statistically significantly associated with APOBEC-3G/F activity, "Containing Premature Stop Codon(s)" if they contained mutations in Gag, Pol, or Env that led to an expected length of less than 95% of known replication-competent strains, or "5′ DEFECT" if they harbored ≥15 nucleotide insertions/deletions compared to the reference genome in the 5′ region upstream of *gag*[30]. If a genome did not contain any of the defects listed above, they were categorized as "genome-intact". Two genomes in this study had incompletely-sequenced 5′ regions but were otherwise intact. Though HIVSeqinR classified these as "inferred intact", we treated both as defective genomes to obtain the highest quality and confidence of intact genomes in this study.

## Viral DNA genome subtyping and recombination analysis

Since viral DNA genomes often contain large deletions, internal inversions, and/or scrambled genomic regions, most viral genomes obtained in this study could not be subtyped using multiple alignment-dependent tools, such as RIP 3.0 and the REGA subtyping tool, that require a continuous query. Here, we developed MOlecular Characterization of HIV-1 (MOCHI) Proviral Subtyping Express 1.0, an R-language-based subtyping tool specific for HIV-1 DNA subtype determination. Our source code is freely available at https://github.com/guineverelee/subtype_express/ (DOI: 10.5281/zenodo.10998392). Briefly, the code attempts to retrieve *gag*, *pol* (PR), *pol* (RT), *pol* (RNaseH), *pol* (INT), *vif, vpr, tat/rev* exon1, *vpu*, *env* (gp120), *env* (gp41), and *nef* regions from a query genome. If present in the query, each viral gene region is then pairwise-aligned with the HIV-1 subtype reference sequences downloadable from the Los Alamos HIV Sequence database. A percentage identity (PID) is then calculated for each query-reference pair. The subtype reference with the highest PID against the query is the viral subtype reported for that region. An inter-subtype recombinant was defined as having more than one viral subtype identified within a single genome, though some calls required manual interpretation (see details in Fig. S1).

## Identifying appropriate IPDA primer and probe locations for the RHSP cohort

To identify amplicon positions that could optimally discriminate proviruses with large deletions in subtypes A1 and D, we applied a sliding window analysis to all HIV-1 DNA sequences containing large deletions collected in this study (n = 495). Each query sequence (FLIP-seq second round amplicon HXB2 638–9638) was mapped to the HXB2 reference genome. Using a 100-base pair (bp) window (representing a hypothetical amplicon), we slid along each viral genome by 10 bp increments and checked for the presence or absence of each window. We then examined all possible pairs of windows along the FLIP-seq amplicon for each query sequence. This allowed us to compute the fraction of all sequences that would be correctly identified as defective (i.e., missing all or part of one or both windows) for each hypothetical amplicon pair.

## Intact proviral quantification

IPDA reactions were prepared as follows: gDNA (a maximum of 700 ng or 7 ng for HIV-1 and RPP30 reactions, respectively) was combined with ddPCR Supermix for Probes (no dUTPs, BIO-RAD catalog # 186-3024), oligonucleotide primers (final concentration 900 nM, Integrated DNA Technologies), probe(s) (final concentration 250 nM, Thermo Fisher Scientific), and nuclease-free water. IPDA-B and IPDA-A1D primer and probe sequences (5′ to 3′) are provided in Table S8. Droplets were prepared using the BIO-RAD QX200 Droplet Generator and cycled at

95 °C for 10 min; 45 cycles of 94 °C for 30 s, 59 °C for 1 min; and 98 °C for 10 min (lid temperature = 105 °C), with a 2.5 °C ramp rate and overnight incubation at 4 °C. Droplets were analyzed on a BIO-RAD QX200 Droplet Reader using QuantaSoft software (BIO-RAD, v1.7.4). A minimum of three technical replicates were performed for each sample and merged before analysis. HIV-negative donor DNA, which represents the double-negative population, is used to guide the location of initial thresholds, which for the IPDA-A1D are normally around x = 2000 and y = 2000. These thresholds are then slightly shifted, if necessary, to accommodate the natural variation in fluorescence amplitudes between samples that occurs due to HIV polymorphism, and to maximize discrimination of positive from negative populations. All technical replicates of the same sample used the same threshold. Positive and negative controls (gDNA from J-Lat cells and an HIV-seronegative individual, respectively) were included in every run. Results were expressed as proviral copies per million CD4+ T-cells, corrected for DNA shearing[16]. When purified HIV-1 PCR amplicons were used as IPDA-A1D templates, results were reported as intact HIV-1 copies per $2 \times 10^4$ total droplets. The IPDA-A1D was verified using DNA extracted from the HIV-1 latency model cell line J-Lat (where each cell harbors an integrated HIV-1 subtype B genome)[23], HIV-infected cell cultures (subtype A1 and D viruses: ARP-2204, -2303, -2386, -2383; obtained from the HIV Reagent Program and cultured in U87 cells), HIV-1 NL4-3 (subtype B virus: ARP-114; obtained from the HIV Reagent Program), CD4+ T-cells from individuals with HIV-1 receiving suppressive ART, and purified near-full-genome PCR amplicons (generated during FLIP-seq).

## Statistical methods

Summary statistics and comparisons between groups (Fisher's exact and Wilcoxon signed-rank tests; Spearman correlation) were performed using GraphPad Prism 9 or R. All tests of significance were two-sided. IPDA-A1D sliding window analyses were performed using R (v4.2.2) and Rstudio (v2023.6.1.524) using R packages profvis (v0.3.7), Iranges (v2.28.0), Biostrings (v2.68.1), ggplot2 (v3.4.3), and dplyr (v1.1.0).

## Reporting summary

Further information on research design is available in the Nature Portfolio Reporting Summary linked to this article.

## Data availability

All raw HIV-1 sequence data that support the findings of this study have been deposited in GenBank. The 607 FLIP-seq sequences are OQ686003–OQ686609, and the QVOA sequences used for HIVSeqinR validation are PP297232–PP297249. All other data supporting the findings of this study are available within the paper and its supplementary information files. Source data are provided with this paper.

## Code availability

Code for the custom software HIVSeqinR for proviral DNA genome-intactness inferences is available at https://github.com/guineverelee/HIVSeqinR/ (DOI: 10.5281/zenodo.10998708). Code for the custom software Molecular Characterization of HIV-1 (MOCHI) Proviral Subtyping Express is available at https://github.com/guineverelee/subtype_express/ (https://doi.org/10.5281/zenodo.10998392).

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

## Acknowledgements

We would like to thank the participants and field teams of the Rakai Health Sciences Program (RHSP) Latent Viral Reservoir study and the participants of HIV reservoir research studies in Vancouver, Canada. We thank the Massachusetts General Hospital Center for Computational & Integrative Biology DNA Core, specifically Nicole Stange-Thomann, Amy Avery, Kristina Belanger, Huajun Wang, and Brian Seed for providing us with the Illumina MiSeq deep sequencing service used in this manuscript. The following reagents were obtained through the NIH HIV Reagent Program, Division of AIDS, NIAID, NIH: J-Lat Full Length Cells (9.2 and 10.6), ARP-9848 and ARP-9849, contributed by Eric Verdin. This work was supported in part by the Division of Intramural Research, NIAID, NIH (ADR, TCQ, SJR); and NIH grants R21AI150398 (GQL), R01AI162221 (GQL), and UM1AI164565 (RBJ). This work was also supported in part by a project grant from the Canadian Institutes of Health Research (CIHR) PJT-159625 (ZLB). MCD and SNG were supported by CIHR CGS-M awards. FHO was supported by a PhD fellowship from the Sub-Saharan African Network for TB/HIV Research Excellence (SANTHE), a DELTAS Africa Initiative (grant # DEL-15-006). The DELTAS Africa Initiative is an independent funding scheme of the African Academy of Sciences (AAS)'s Alliance for Accelerating Excellence in Science in Africa (AESA) and supported by the New Partnership for Africa's Development Planning and Coordinating Agency (NEPAD Agency) with funding from the Wellcome Trust (grant # 107752/Z/15/Z) and the UK government. NNK was supported by a CIHR Vanier Canada Graduate Scholarship. ZLB was supported by a Scholar Award from the Michael Smith Foundation for Health Research. JLP was supported by the Canada Research Chairs Program (Canadian Institutes of Health Science 950 – 233211).

## Author contributions

The work presented here was carried out in collaboration between all authors. The study was conceptualized and designed by G.Q.L., A.D.R., and J.L.P., with Z.L.B. contributing to IPDA-A1D design. PBMC samples and clinical/demographic data were collected by R.M.G., J.K., T.K., P.B., S.J., S.T., A.A., S.J.R., T.C.Q., A.D.R., and J.L.P. HIV-1 genotyping laboratory work was done by G.Q.L. and P.K. Sliding window analysis was performed by S.N.G. and supervised by G.Q.L. Subtype B samples and associated IPDA-B data were provided by N.N.K., F.H.O., M.C.D., and Z.L.B. IPDA-A1D primers and probes were designed by G.Q.L. with N.N.K. and Z.L.B. IPDA-A1D validation laboratory work was done by S.N.G. and D.C.C. Results were analyzed by G.Q.L., P.K., D.C.C., S.N.G., Z.C., R.B.J., Z.L.B. G.Q.L. performed the bioinformatics analysis using HIVSeqinR and MOCHI Proviral Subtyping Express. G.Q.L., Z.L.B., A.D.R., and J.L.P. wrote the manuscript; all authors contributed to and approved the manuscript. G.Q.L., Z.L.B., A.D.R., and J.L.P. supervised the study execution.

## Competing interests

The authors declare no competing interests.
