## [Peer Review File · Nature Communications]

HIV-1 subtype A1, D, and recombinant proviral genome landscapes during long-term suppressive therapyREVIEWER COMMENTS

Reviewer #1 (Remarks to the Author):

The authors analyzed the details of HIV-1 provirus in peripheral blood of individuals infected with subtype A1, D, and some recombinant viruses. They found that there are both similarities and differences between subtype A1 or D, and subtype B, which is the most extensively characterized subtype. There were some important findings, such as subtype A specific variation in env probe in IPDA assay. However, study is a little bit descriptive and the number of infected individuals analyzed was relatively small as the authors mentioned.

I'd like to add some specific comments as below.

1. Subtype A specific variation in env probe in IPDA assay

As I mentioned above, this is a valuable message. The data indicated that we will need to take this diversity into account when we perform IPDA assay in the country that contain HIV subtype different from subtype B. It would enhance the value of this study if the authors show the alternative probe that is useful in subtype A virus.

2. It would be more informative for the readers if the authors describe the details how they define intact provirus.

3. They sometimes used CD4 + cells, but other times they used CD4+ T cells.

Reviewer #2 (Remarks to the Author):

Lee, et al. analyzed near full-length HIV proviral sequences in 23 PLWH with subtypes A1, D, and A1/D to determine the percent intact, the types of defects, the degree of clonal expansion, and their adaptability to the IPDA. This study is an important contribution to the HIV cure field. The majority of cure-related research is performed using samples collected from men with subtype B infection. However, most PLWH in the world are women with subtypes other than B. Much credit should be given to this group for addressing a very important gap in the cure field. The authors made the surprising finding that the recombinant forms of A1/D did not have breakpoints in common. Each recombinant virus contained unique breakpoints demonstrating that co-infection with subtype A1 and D is frequent and that new recombinant forms are being generated very often. They also found that the percent of intact proviruses that persist on long-term ART is similar to what was reported for subtype B but that the primer/probe binding sites for the IPDA are more diverse, making adaptation of this screening tool a challenge. This study will be an important contribution to the literature but a few modifications to the text of the paper are needed.

Methods:

- The authors should include the target region for measuring HIV DNA copies in the text of the methods. The target is included in Table S2 but it would be helpful to include it in the text as well.
- A column-based kit for gDNA extraction was used which runs the risk of cutting the provirus. As a result, there may be proviruses that were missed due to random sheering by the column. This limitation should be mentioned or methods to overcome this limitation described.
- There is the assumption that only one replication-competent virus was induced in the QVOA wells. It cannot be completely ruled out that some wells included outgrowth from more than 1 infected cell and a product of recombination occurred in the outgrowth. More specifically, if a donor was dual-infected where two infected cells were plated (one as A1 and D, from two separate events), the outgrowth virus would look as a A1/D recombinant. The authors should address this possibility or explain how their methods avoided this possibility.
- What programs were used to determine subtype and the breakpoints? Does the tiled approach normalize for the length of the region? Depending on where breakpoints occur and the length, could a particular crossover be missed?
- FLIPS was shown to amplify proviruses with large internal deletions more efficiently than intact

proviruses. If this limitation of FLIPS was overcome, please add the changes to the protocol to address this issue. If not, then state the limitation and reference the paper that describes it.

Results:

- Based on recent reports from the PANGEA consortium, do the subtypes/recombinants reflect the Ugandan landscape?

Discussion:

- A common A/D URF observed in Uganda has breakpoints at HXB2 6760 +/-250bp and 8710 +/-100bp. Do the A/D recombinant in this study share these breakpoints?
- The study analyzed 600+ proviral sequences across 23 donors with two subtypes and with recombinant forms. While mechanistically there doesn't appear to be a clear reason why there would be more intact proviruses in any of these subtypes, the study may not have the power to make this conclusion. Therefore, it would be best to soften the conclusions to state that there is no evidence that the % intact proviruses is different than subtype B, but more studies are needed.
- The authors should include the implications of the finding that each recombinant A1/D was unique. Can this data be used to inform the number of co-infections in the region? What is the impact of this high rate of co-infection? Is it possible that any of the individuals in this study were co-infected and that the most fit recombinant became dominant in the population?

Other comments:

- Add IUPM to Table S2

Reviewer #3 (Remarks to the Author):

The work performed by Lee and colleagues evaluates the proviral landscape of HIV- subtypes A1, D and recombinant genomes in PLWH from Rakai, Uganda.

The work was done in depth, but apart from the fact that the work was done in non-B subtype, the novelty is limited. As mentioned briefly in the discussion, It is impossible to draw conclusions about non-significant differences for intact 20/234 for subtype D and 7/249 for A1/D recombination, which can certainly be explained by too small numbers. The paper and reporting could be significantly improved.

Major comments

- As non-subtype B samples pose a significant challenge, correct primer design is required for adequate near full-length HIV-1 genome amplification to prevent drop outs. The authors do not state exactly how they validated their primers (line #82-83) used for FLIP-Seq. Please elaborate on how validation was performed (did the authors use reference sequences, which criteria for accepting a primer design).
- The method section lacks any information on how exactly the HIV-1 genomes were reconstructed using de novo assembly from short-read NGS data (line #84-85). It does not mention the quality and trimming steps, assembly software or parameters used for the assembly process. This is a crucial piece of information to allow replicability by others that should be included (eg GitHub repository)
- No mention on which algorithms were used to perform the subtyping of proviral genomes (for both near full-length or subgenomic pol/gp41 regions). Given the subject of this paper, this is relevant information that should be included. Did the authors resort to using publicly available tools (eg. Rega HIV subtyping tool, COMET, HIVDB subtyping program) or devise their own strategy (blast-based, phylogeny based. Please elaborate. Note that the selected strategy (and how up to date the included reference database) might also influence discovery of circulating recombinant forms (line #124-125).
- On line #85-87, the authors state the following "manually inspected for inference accuracy due to potential subtype specific bioinformatic parameters" but make no mention if indeed they observed any subtype related parameters that might influence intactness calling. This should be

part of the discussion as it might add another challenge for interpreting non subtype B proviruses.

- If I understand correctly, the results shown in Figure 4 are only based on the presence of regions assessed by IPDA (psi and/or env) and do not take into account primer/probe diversity (mismatches) except for the exclusion of hypermutated genomes. Are the calculated sensitivities and specificities not flawed? I suspect some genomes might perhaps produce no amplicons or don't allow for probe hybridization during ddPCR due to mismatches? Take for instance the psi probes in D, A1D.
- I was interested in the proviral class of viruses that would be considered intact by IPDA but are defective based on their near full-length genome. Interestingly, from Figure 4 and Supplementary Table 7 I see that 9/19 (47%) are labelled 5DEFECT. Could you provide more information on why these viruses are 5' defective but still work with IPDA? Isn't this odd as the IPDA is designed on conserved regions (for subtype B) but seems not to work here? Can this be explained by the subtype?

Minor comments

- figure 1 is initially not discussed, but only mentioned, which complicates the flow of the paper. Legend figure 1 not clear.
- Figure 2: first sequence seems recombinant (with PR), which only becomes clear in the discussion, is not discussed in the results
- Mistake in column header of Supplemental Table 7: 'hypermuted' should be 'hypermutated'.
- Mistake in column header of Supplemental Table 2: 'copoies' should be 'copies'.
- Authors do no mention which kit was used to perform the library preparation of their HIV-1 amplicons. Please include.
- On line #84, it appears that the authors used a 2x150bp run, and the notation of 150x2bp is likely an error.
- The authors should add a Supplemental table listing all primers and probes of the assays used in this study for replicability.
- In Figure 1, the legend shows a variety of HIV-1 genome classes without any explanation or definition (some can be inferred, but what's this 'Scramble' class, difference between Intact and Inferred Intact). Are genomes with Inferred Intact status added to the count of total Intact genomes?
- The strategy used to evaluate the proviral class and intactness of HIV proviral genomes is mentioned HIVSeqinR code (line #87). However a clear description of the exact definitions for each class (intact, 5' leader defect) is missing. Maybe the GitHub documentation could be updated and offer a short description.
- The authors mention also performing near full-length sequencing on QVOA wells (line #88). Did they find identical overlaps between FLIP-Seq proviruses and QVOA genomes?
- Will the 607 HIV-1 proviral genomes from this study be made publicly available, if possible, via GenBank or the HIV Proviral Sequence Database. This will inform other researchers to assess primer/probes of their assays and validate whether they would work on non subtype B viruses.
- In Supplementary Figure 2, for the psi forward primer an asterisks is shown for the D subtypes but no variation is visible. Mistake?
- On line #167, the authors state the following "whereas 5% (28/547) of defective genomes contained IPDA binding sites and were not hypermutated". Am I correct at interpreting that these are non-hypermutated defective genomes? If so the numbers from Figure 4 and Supplementary Table 7 don't add up as I only count 19 psi+/env+ defective NFL genomes.

Responses to reviewer comments

Reviewer #1 (Remarks to the Author):

The authors analyzed the details of HIV-1 provirus in peripheral blood of individuals infected with subtype A1, D, and some recombinant viruses. They found that there are both similarities and differences between subtype A1 or D, and subtype B, which is the most extensively characterized subtype. There were some important findings, such as subtype A specific variation in env probe in IPDA assay. However, study is a little bit descriptive and the number of infected individuals analyzed was relatively small as the authors mentioned.

I'd like to add some specific comments as below.

1. Subtype A specific variation in env probe in IPDA assay

As I mentioned above, this is a valuable message. The data indicated that we will need to take this diversity into account when we perform IPDA assay in the country that contain HIV subtype different from subtype B. It would enhance the value of this study if the authors show the alternative probe that is useful in subtype A virus.

We thank reviewer #1 for their positive feedback. The revised manuscript now details our selection and validation of IPDA primers/probes designed for the diverse subtype A1 and D HIV-1 sequences observed in Rakai. This information is detailed in the methods sections "Identifying appropriate IPDA primer and probe locations for the RHSP cohort (lines 376-385)," "Intact proviral quantification (lines 387-407)," and results sections "IPDA target assessment: genomic location (lines 139-158)," "Adaptation of the IPDA for use in the RHSP cohort (lines 160-182)," and "IPDA-A1D validation (lines 184-217)" as well as a new Figure 4, Table S8 and Figures S5-S7.

2. It would be more informative for the readers if the authors describe the details how they define intact provirus.

Our revised manuscript now includes a new methods section "Bioinformatics determination of viral DNA genome intactness" (lines 337-357), in which we describe the definition of genome-intactness. We also now include a new Supplementary section, "Validation of HIVSeqinR for HIV subtypes A1 and D", found on pages 46-54 of the new supplementary document, which contains new Figures S10-S15 describing our validation of HIVSeqinR for use in subtype A1/D.

3. They sometimes used CD4 + cells, but other times they used CD4+ T cells.

We thank reviewer 1 for identifying this inconsistency. We have now corrected and unify all mentions of CD4+ to "CD4+ T-cells".

Reviewer #2 (Remarks to the Author):

Lee, et al. analyzed near full-length HIV proviral sequences in 23 PLWH with subtypes A1, D, and A1/D to determine the percent intact, the types of defects, the degree of clonal expansion, and their adaptability to the IPDA. This study is an important contribution to the HIV cure field. The majority of cure-related research is performed using samples collected from men with subtype B infection. However, most PLWH in the world are women with subtypes other than B. Much credit should be given to this group for addressing a very important gap in the cure field. The authors made the surprising finding that the recombinant forms of A1/D did not have breakpoints in common. Each recombinant virus contained unique breakpoints demonstrating that co-infection with subtype A1 and D is frequent and that new recombinant forms are

being generated very often. They also found that the percent of intact proviruses that persist on long-term ART is similar to what was reported for subtype B but that the primer/probe binding sites for the IPDA are more diverse, making adaptation of this screening tool a challenge. This study will be an important contribution to the literature but a few modifications to the text of the paper are needed.

We thank reviewer 2 for the positive and constructive comments.

Methods:

- The authors should include the target region for measuring HIV DNA copies in the text of the methods. The target is included in Table S2 but it would be helpful to include it in the text as well.

The target genome range for measuring total HIV DNA copies is HXB2 684-810 (Malnati 2008 Nat Protoc). This information is now added to the Methods section (line 312).

- A column-based kit for gDNA extraction was used which runs the risk of cutting the provirus. As a result, there may be proviruses that were missed due to random shearing by the column. This limitation should be mentioned or methods to overcome this limitation described.

We acknowledge this limitation. The discussion now states (line 268-271). “FLIP-seq uses a column-based kit for genomic DNA extraction, which is prone to genomic DNA shearing and therefore there is a possibility that not all full-length viral genomes in the samples were successfully captured (Lee *Viruses* 2021).”

- There is the assumption that only one replication-competent virus was induced in the QVOA wells. It cannot be completely ruled out that some wells included outgrowth from more than 1 infected cell and a product of recombination occurred in the outgrowth. More specifically, if a donor was dual-infected where two infected cells were plated (one as A1 and D, from two separate events), the outgrowth virus would look as a A1/D recombinant. The authors should address this possibility or explain how their methods avoided this possibility.

We thank the reviewer for this comment. We would like to clarify that the sequencing of the QVOA wells targeting *pol* (HXB2 2723–3225) and *gp41* (HXB2 7938–8256) was only used to guide the selection of study participants, with the goal of representing individuals with subtype A1, D and recombinants in the study.

Nevertheless, we acknowledge that QVOA wells can contain multiple infected cells and that recombination can occur *in vitro* during culture. However, subgenomic Illumina deep sequencing (*pol* and *gp41*) can distinguish between wells containing one versus multiple infected cells. Specifically, wells containing two non-clonal, replication-competent viruses will show two distinct viral species in *pol* and/or *gp41*. We may also observe less prominent variants that are the result of within-assay evolution; these variants are removed during data cleaning, as previously described in Poon et al 2018 (reference 30 in the revised manuscript). Only wells containing a single variant at both the *pol* and *gp41* regions were used for the original subtype determination. We have now clarified this in the Methods section (Line 301-306):

“The participants examined here were initially selected based on (i) the availability of frozen rCD4 cells and (ii) viral subtype data, to obtain representation of subtype A1, D, and A1/D recombinants. This initial viral subtyping for sample selection was performed by Illumina deep-sequencing of *pol* (HXB2 2723–3225) and *gp41* (HXB2 7938–8256) sequences from p24-positive QVOA wells that contained a single dominant viral sequence in both regions (Poon *et al* 2018).”

- What programs were used to determine subtype and the breakpoints? Does the tiled approach normalize for the length of the region? Depending on where breakpoints occur and the length, could a particular crossover be missed?

An in-house bioinformatics algorithm, MOCHI Proviral Subtyping Express 1.0, was used to determine viral subtypes, followed by result validation by RIP 3.0. We developed a new algorithm because existing programs such as RIP 3.0 and/or REGA cannot accommodate proviral genomes with large deletions or inversions that dominate during ART (RIP 3.0 and REGA can only process continuous (i.e. non-truncated) HIV-1 sequences in the sense strand).

In this revision, we have made our in-house code publicly available in GitHub (https://github.com/guineverelee/subtype_express/).

Briefly, the code takes a query viral genome (FLIP-seq HXB2 638-9638 in this study) and tries to retrieve *gag*, *pol* (PR), *pol* (RT), *pol* (RNaseH), *pol* (INT), *vif*, *vpr*, *tat/rev* exon1, *vpu*, *env* (gp120), *env* (gp41) and *nef* from the query genome. Each of the viral genes, if present in the query, are then pairwise-aligned with the HIV subtype reference sequences downloadable from the Los Alamos HIV Sequence databases. Each query-reference sequence pair yields a percentage identity (PID) score. The subtype reference with the highest PID against the query is the viral subtype reported. For example, referring to donor 2's subtyping result shown in the new Supplementary Figure S1 (page 5 of new supplementary document), all the genomes derived this donor had a *gp41* sequence that most closely matched the subtype D reference, whereas the neighboring *nef* sequence most closely matched the subtype A1 reference. It is important to note however that we are not “pinpointing” recombination breakpoints down to the exact coordinates, which would require more specialized analyses that will require development of new bioinformatics tools because current tools will not handle proviral genomes with gross defects such as large deletions and/or internal inversions. Rather, MOCHI's objective was to assign each viral gene to a subtype, and look for the presence of multiple viral subtypes within a viral genome. In the case of donor 2, all genomes were subtype D between *gag* to *gp41*, but *nef* was always subtype A1. Even though we did not pinpoint the specific breakpoint, our analysis nevertheless indicates that it is near the end of *gp41* and the beginning of *nef*, and that all viral genomes from this individual shared the same general breakpoint.

It is also important to note that the tiles in Supplementary Figure S1 do not reflect the length of each indicated gene region. Rather, these plots were designed to allow users to quickly visualize whether a given viral genome is comprised of multiple subtypes, and if so, which ones.

We have added a new Methods section titled “Viral DNA genome subtyping and recombination analysis” containing this information. (lines 359-375). A link to MOCHI's GitHub source code is included in-text.

- FLIPS was shown to amplify proviruses with large internal deletions more efficiently than intact proviruses. If this limitation of FLIPS was overcome, please add the changes to the protocol to address this issue. If not, then state the limitation and reference the paper that describes it.

We used the original FLIP-seq (Lee JCI 2019, Lee Nat Comm 2019) with no modifications. The discussion of our manuscript now acknowledges that FLIP-seq is biased towards amplifying proviruses with large internal deletions, referencing White 2022 PLoS Path (lines 271-272). Due to this limitation, we do not use the number of intact genomes per sample to quantify reservoir sizes, but instead focus on characterizing viral DNA genetic diversity in this cohort (lines 272-274).

Results:

- Based on recent reports from the PANGAEA consortium, do the subtypes/recombinants reflect the Ugandan landscape?

Yes. Our subtyping/recombinant results cohort are consistent with data from the PANGEA consortium (Pillay Lancet Infect Dis. 2016, Abeler-Dörner Curr Opin HIV AIDS 2019, Capoferri 2020 AIDS Res Hum Retroviruses, Grant Retrovirology 2022) and data from the UARTO cohort (Lee AIDS 2017). This information, and these references have been added to our discussion (lines 236-238).

Discussion:

- A common A/D URF observed in Uganda has breakpoints at HXB2 6760 +/-250bp and 8710 +/-100bp. Do the A/D recombinant in this study share these breakpoints?

As mentioned above, existing recombination analysis tools for HIV require a continuous query sequence without major truncations and inversions, and thus cannot be used for this dataset. The subtyping tool (MOCHI) that we developed to address this does not output coordinate-specific breakpoints. We are therefore unable to conclusively identify whether any of our donors had this common recombinant. However, we did observe that Donor 2 (the example given above) recombinant pattern was consistent with the breakpoints mentioned by the reviewer. No other donors had this pattern.

- The study analyzed 600+ proviral sequences across 23 donors with two subtypes and with recombinant forms. While mechanistically there doesn't appear to be a clear reason why there would be more intact proviruses in any of these subtypes, the study may not have the power to make this conclusion. Therefore, it would be best to soften the conclusions to state that there is no evidence that the % intact proviruses is different than subtype B, but more studies are needed.

We agree with the reviewer that our study conclusion is limited by our sampling depth. We now acknowledge the limited sampling in the discussion (lines 274-277) have softened our conclusion as suggested (lines 279-280).

- The authors should include the implications of the finding that each recombinant A1/D was unique. Can this data be used to inform the number of co-infections in the region? What is the impact of this high rate of co-infection? Is it possible that any of the individuals in this study were co-infected and that the most fit recombinant became dominant in the population?

We have added these lines to our discussion section, “[O]ur finding that each recombinant A1/D genome was unique is suggestive of a high frequency of subtype A1 and D dual infection in the region. However, since we intentionally selected participants based on their initial subtyping results, and purposely included individuals with recombinant infections, the subtype distribution in this study is not necessarily representative and therefore we cannot use our observations to estimate the level of dual infection in the region. Our findings are nevertheless consistent with our previous observations from an independent cohort, where inter-subtype recombinants with unique breakpoints (as determined by the Los Alamos Recombinant Identification Program, RIP) made up 46% of HIV-1 infections in Mbarara, Uganda between 2005-2010 (Lee AIDS 2017). Future population-level and longitudinal studies will be required to address the prevalence and viral evolution patterns of multi-subtype infections leading to recombination.” (lines 246-256)

Other comments:

- Add IUPM to Table S2

We have added IUPM data to the first column of this table, now numbered Supplementary Table S2.

Reviewer #3 (Remarks to the Author):

The work performed by Lee and colleagues evaluates the proviral landscape of HIV- subtypes A1, D and recombinant genomes in PLWH from Rakai, Uganda.

The work was done in depth, but apart from the fact that the work was done in non-B subtype, the novelty is limited. As mentioned briefly in the discussion, It is impossible to draw conclusions about non-significant differences for intact 20/234 for subtype D and 7/249 for A1/D recombination, which can certainly be explained by too small numbers. The paper and reporting could be significantly improved.

We thank Reviewer 3 for the feedback. We hope that the changes made to the manuscript, described below, alleviate the reviewer's remaining concerns.

Major comments

- As non-subtype B samples pose a significant challenge, correct primer design is required for adequate near full-length HIV-1 genome amplification to prevent drop outs. The authors do not state exactly how they validated their primers (line #82-83) used for FLIP-Seq. Please elaborate on how validation was performed (did the authors use reference sequences, which criteria for accepting a primer design).

We appreciate this comment. We have detailed this validation in a new Supplementary section, “ddPCR and FLIP-seq primer/probe validation”, which is referenced in the Methods section (line 315-316), and can be found on pages 42-45 of the supplementary document.

- The method section lacks any information on how exactly the HIV-1 genomes were reconstructed using *de novo* assembly from short-read NGS data (line #84-85). It does not mention the quality and trimming steps, assembly software or parameters used for the assembly process. This is a crucial piece of information to allow replicability by others that should be included (eg GitHub repository)

The HIV genomes were derived from sequencing FLIP-seq SGA amplicons using Illumina MiSeq (150bp \times 2), followed by processing the resulting small reads using a proprietary in-house *de novo* assembly pipeline offered as a part of the Massachusetts General Hospital Center for Computational and Integrative Biology (CCIB) DNA core service.

We have amended our methods section to include additional details on sequence quality control as follows (lines 316-326): “SGA PCR amplicons were subjected to library preparation using the Illumina Nextera XT DNA Library Preparation Kit (modified to replace the transposon cleavage step with physical fragmentation by sonication resulting in fragment lengths averaging 250 base pairs), and sequenced on an Illumina MiSeq (2 \times 150bp) at the Massachusetts General Hospital (MGH) Center for Computational and Integrative Biology (CCIB) DNA core. Resulting short reads (average 117,000 per amplicon) were *de novo* assembled using the DNA core *de novo* assembler UltraCycler v1.0 (proprietary). The average coverage per position was 2,520 reads. At each nucleotide position, the majority base was taken as the final identity of the position. To monitor PCR and/or sequencing errors, the frequency of mismatches at each nucleotide position was calculated: Over 99.6% of base positions contained fewer than 5% mismatches to the final assigned base.”

When we developed FLIP-seq (Lee JCI 2017), we compared near-full-genome Sanger sequencing with this Illumina method, which resulted in 100% concordance in base calls. As part of this previous study, we also processed raw FASTQ files provided by the DNA core using SPAdes *de novo* assembler (default settings) to reconstruct the contigs, which resulted in zero mismatches against the DNA core output.

- No mention on which algorithms were used to perform the subtyping of proviral genomes (for both near full-length or subgenomic pol/gp41 regions). Given the subject of this paper, this is relevant information that should be included. Did the authors resort to using publicly available tools (eg. Rega HIV subtyping

tool, COMET, HIVDB subtyping program) or devise their own strategy (blast-based, phylogeny based. Please elaborate. Note that the selected strategy (and how up to date the included reference database) might also influence discovery of circulating recombinant forms (line #124-125).

We appreciate this comment, which reviewer 2 also raised. As detailed in our response to reviewer 2, we developed an in-house bioinformatics algorithm, MOCHI Proviral Subtyping Express 1.0, to determine viral subtypes because existing programs such as RIP 3.0 and/or REGA cannot accommodate HIV sequences with gross defects. Our code is available at https://github.com/guineverelee/subtype_express/.

- On line #85-87, the authors state the following “manually inspected for inference accuracy due to potential subtype specific bioinformatic parameters” but make no mention if indeed they observed any subtype related parameters that might influence intactness calling. This should be part of the discussion as it might add another challenge for interpreting non subtype B proviruses.

Our revised manuscript now contains a Supplementary section, “Validation of HIVSeqinR for HIV subtypes A1 and D,” that describes the four subtype-sensitive parameters associated with HIVSeqinR, and our efforts to test each of them for appropriateness on subtypes A1 and D. This new supplementary section is now referenced in the methods (Line 339-341) and can be found on pages 46-54 of the supplementary document.

- If I understand correctly, the results shown in Figure 4 are only based on the presence of regions assessed by IPDA (psi and/or env) and do not take into account primer/probe diversity (mismatches) except for the exclusion of hypermutated genomes. Are the calculated sensitivities and specificities not flawed? I suspect some genomes might perhaps produce no amplicons or don't allow for probe hybridization during ddPCR due to mismatches? Take for instance the psi probes in D, A1D.

Yes, sensitivity and specificity estimates in Figure 4 (now Supplementary Figure S3) were solely based on the location of the primers and probes and did not take into consideration sequence diversity. We have revised the text to clearly indicate that these estimates consider primer/probe location only (lines 149-153).

More importantly, and as detailed in our response to reviewer 1, the revised manuscript contains a design and validation of an IPDA assay adapted for subtypes A1 and D. This information is detailed in the methods sections “Identifying appropriate IPDA primer and probe locations for the RHSP cohort (lines 376-385),” “Intact proviral quantification (lines 387-407),” and results sections “IPDA target assessment: genomic location (lines 139-158),” “Adaptation of the IPDA for use in the RHSP cohort (lines 160-182),” and “IPDA-A1D validation” (lines 184-217) as well as a new Figure 4, Table S8 and Figures S5-S7.

- I was interested in the proviral class of viruses that would be considered intact by IPDA but are defective based on their near full-length genome. Interestingly, from Figure 4 and Supplementary Table 7 I see that 9/19 (47%) are labelled 5DEFECT. Could you provide more information on why these viruses are 5' defective but still work with IPDA? Isn't this odd as the IPDA is designed on conserved regions (for subtype B) but seems not to work here? Can this be explained by the subtype?

HIVSeqinR's definition of 5DEFECT is “the presence of large (≥ 15 nucleotides) insertions or deletions in the 5' untranslated region of the genome between HXB2 coordinates 638-789, which stretches from start of PCR amplicon to the *gag*”. The IPDA primers/probes in this region are located between HXB2 692-797 only. Proviruses with mutations between 638-691, or with smaller mutations located in between the IPDA primer and probe binding sites can therefore still be detected by the IPDA.

Minor comments

- figure 1 is initially not discussed, but only mentioned, which complicates the flow of the paper. Legend figure 1 not clear.

We have edited the flow of the main text (lines 91-123) and have made the original Figure 1 into Figure 2. The legend has also been edited for clarity.

- Figure 2: first sequence seems recombinant (with PR), which only becomes clear in the discussion, is not discussed in the results

PR from this donor best matched subtype 01_AE consensus (green). But, further analysis of subtype 01_AE consensus using the Los Alamos HIV Sequence Database Recombination Identification Program (RIP) revealed that the 5' half of this consensus sequence (until HXB2 coordinate 5387) mapped to pure consensus A1. Therefore, this donor (donor 17) was categorized as having subtype A1 HIV-1 in the rest of the manuscript. The figure legend (formerly Figure 2, now Figure 1) clarifies this point.

- Mistake in column header of Supplemental Table 7: 'hypermuted' should be 'hypermuted'.

We have changed "hypermuted" to "hypermuted" in Supplementary Table S7.

- Mistake in column header of Supplemental Table 2: 'copoies' should be 'copies'.

We have now corrected this error in Supplementary Table S2.

- Authors do no mention which kit was used to perform the library preparation of their HIV-1 amplicons. Please include.

The Illumina MiSeq library preparation and sequencing was performed by the Massachusetts General Hospital Center for Computational and Integrative Biology (CCIB) DNA core. They used a modified Nextera XT protocol by replacing the transposon cleavage step with physical fragmentation by sonication resulting in fragment lengths averaging 250 base pairs. This information has been added on lines 316-321 in the Methods section.

- On line #84, it appears that the authors used a 2x150bp run, and the notation of 150x2bp is likely an error.

Thank-you for catching our error, which we have now corrected.

- The authors should add a Supplemental table listing all primers and probes of the assays used in this study for replicability.

All primer and probe sequences used in this study are now indicated in Supplementary Table S8 (IPDA-B and IPDA-A1D) and Supplementary Table S9 (ddPCR and FLIP-seq validation for subtype A1 and D)

- In Figure 1, the legend shows a variety of HIV-1 genome classes without any explanation or definition (some can be inferred, but what's this 'Scramble' class, difference between Intact and Inferred Intact). Are genomes with Inferred Intact status added to the count of total Intact genomes?

The revised methods section now features a brief description of these categories on lines 342-354, and we have also added these definitions to the figure legend (was Figure 1, now Figure 2). The exact definitions of proviral class and intactness can be found in Lee Viruses 2021. The "scramble" class identifies genomes where one or more HIV genes are in the incorrect order, but without inversion (e.g. if protease is found 5' upstream of *gag*). "Inferred intact" includes any genome with a missing 5' primer binding site but is

otherwise intact, which suggests a possible sequencing artifact. Only two inferred intact sequences were in this dataset of 607 genomes, and the IPDA psi+ site was not captured in either genome. To ensure the highest quality and confidence of genome-intactness, both sequences were excluded from our analyses of genome-intact sequences.

- The strategy used to evaluate the proviral class and intactness of HIV proviral genomes is mentioned HIVSeqinR code (line #87). However a clear description of the exact definitions for each class (intact, 5' leader defect) is missing. Maybe the GitHub documentation could be updated and offer a short description.

Please see response above; exact definitions of proviral class and intactness can be found in Lee Viruses 2021. We have also included a brief description of the definitions on lines 342-354.

- The authors mention also performing near full-length sequencing on QVOA wells (line #88). Did they find identical overlaps between FLIP-Seq proviruses and QVOA genomes?

The purpose of near full-length sequencing on QVOA wells was to validate the HIVSeqinR pipeline for subtype A1 and D HIV-1 (see Supplementary section "Validation of HIVSeqinR for HIV subtypes A1 and D" found on pages 46-54 of the new supplementary document). It was performed on 18 donors, and only 4/18 overlapped with the 23 donors from which the 607 viral DNA genomes were obtained from, due to a lack of sample availability. We did not see identical overlaps between FLIP-seq proviruses and QVOA-derived genomes after constructing a phylogenetic tree and running a percentage identity analysis.

- Will the 607 HIV-1 proviral genomes from this study be made publicly available, if possible, via GenBank or the HIV Proviral Sequence Database. This will inform other researchers to assess primer/probes of their assays and validate whether they would work on non subtype B viruses.

The GenBank accession numbers of these 607 genomes are OQ686003 – OQ686609. We have now included this information in the methods (line 335) and in the data availability statement (lines 416-422).

- In Supplementary Figure 2, for the psi forward primer an asterisks is shown for the D subtypes but no variation is visible. Mistake?

Thank you for spotting this error. We have removed the asterisk. Supplementary Figure 2 is now Supplementary Figure S4 in this revision.

- On line #167, the authors state the following “whereas 5% (28/547) of defective genomes contained IPDA binding sites and were not hypermutated”. Am I correct at interpreting that these are non-hypermutated defective genomes? If so the numbers from Figure 4 and Supplementary Table 7 don’t add up as I only count 19 psi+/env+ defective NFL genomes.

We have clarified the text and edited the values to reflect the counts as shown in Figure 4 (now Supplementary Table S3) and Supplementary Table 7. The text now reads, “only 3% (19/575) of defective genomes were observed to contain both of the IPDA-B binding sites but were not hypermutated...” (lines 151-152).

REVIEWER COMMENTS

Reviewer #1 (Remarks to the Author):

The authors carefully and substantially revised manuscript according to the reviewers' comments. They have done adaptation of the IPDA for use in the RHSP cohort in revised paper. Further, the authors carefully validated the efficiency of the IPDA-A1D. They applied the IPDA-A1D to samples derived from participants in the RHSP cohort. IPDA-B failed to detect intact provirus in all cases (n=4), but the IPDA-A1D rescued the detection of intact proviruses.

I agree with the authors that optimization of IPDA based on actual sequence data would give us more valuable and practical advantage for the clinical application in near future.

As authors mentioned, HIV-1 infection is more evident in Africa, but previous studies so far focused more on HIV-1 subtype B, which is the most common in western countries but not in Africa.

HIV-1 provirus characterization in non-B subtypes has not been performed as that in subtype B. Thus, the data presented in this study would be valuable information to think about how HIV-1 infection in Africa would be regulated to reduce HIV burden there.

The revised manuscript is much more improved and informative than initial version. I have no additional comments.

Reviewer #2 (Remarks to the Author):

All of my previous concerns and suggestions were addressed thoroughly. I have no additional questions or concerns. In my view, the manuscript fills an important gap in the field related to expanding the gold-standard approach for characterizing the HIV reservoir in people living with HIV to people are not men with subtype B infection. I commend the authors for including women and people with other subtypes, including recombinant forms, in studies of HIV persistence and cure-related research. The finding that co-infections leading to diverse recombinant forms of HIV is a frequent occurrence in the world is an important contribution to the field.

Reviewer #3 (Remarks to the Author):

The authors have made significant improvements to the presented manuscript. We commend them for their efforts and sharing more detailed overviews on their methodology and analysis (MOCHI, primer/probe designs and intactness parameters) as requested. Some remarks remain upon revision:

Major remarks

- I would contest the following statement on lines 69-72 and ask to rephrase:

"As such, the suitability of the primer/probe locations and sequences for use in Africa, which is affected almost exclusively by non-B subtypes, remains a major knowledge gap and poses a barrier to cure research in the regions where HIV-1 disease burden is the heaviest globally."

While proviral reservoir sequences are indeed understudied, the suitability of IPDA primer/probe designs targeting subtypes A and D have been studied before, for example by Cassidy et al. (<https://www.ncbi.nlm.nih.gov/pmc/articles/PMC8786636/>: cross-subtype IPDA) (referenced later in line 260). Please acknowledge other efforts

(<https://www.ncbi.nlm.nih.gov/pmc/articles/PMC8786636/>,
<https://link.springer.com/article/10.1186/s12985-024-02300-6>)

- The statement on lines 126-128 can be misleading. While the authors refer to previous work (using MIP-Seq data with linked IS and provirus) to prove their hypothesis on identical NFL sequences being derived from clonal expansion, this idea doesn't hold when applied to FLIP-Seq data. First of all, the sole proof for a clone is having a linked integration site (which FLIP-Seq doesn't give) and is missing from these datasets. Next, others have shown (<https://doi.org/10.1371/journal.ppat.1005689>) that identical subgenomic sequences (even being NFL) are not direct proof of identical full-length HIV-1 genomes (aka clonal infected cells). While the statement might hold true when applied to a cohort of individuals who initiated therapy late after HIV-1 acquisition, it certainly would not stand when applied to samples from PWH initiating therapy immediately after HIV-1 acquisition due to limited proviral diversity. I suggest to rephrase and clearly mention the limitations of this analysis, and not (misleadingly) state this as a general fact.

- Concerning the quantification of intactness: the authors report their median total but don't really say anything about the size of the intact reservoir. They do say on line 216 that they were able to obtain intact values in all samples with the new design, but I am curious about the results. The examples in Figure 4 are also quite high, according to the fact that those people are on therapy: donor 7  140 intact/million

donor 20 250 intact/million. it would be a nice addition if the authors reported total and intact for each donor. In this way the reader/reviewer can see for each patient whether it makes sense that they do- or do not- obtain intact sequences with FLIPS.

- In the methods the authors report that they use the quantasoft software for analysis, but how do they do the thresholding? Do they let the software set thresholds or do they adjust them for each sample? Eg You can see from the plots that they are slightly different for the 3 donors.

- For donor 20 they say that IPDA-A1D partially captures the SNPs, does this mean that they think they can only quantify part of the intact reservoir? can they test this with patient-specific degenerate primer/probe sets?

- The use of degenerate primer/probe sets results in less good separation between positive and negative partitions: the authors mention that in the part in which they test the probes on JLat cells. Applying this assay to samples where there are even more mismatches than the ones they cover can result in no separation at all. Can the authors further comment on that.

in the methods the authors report that they use the quantasoft software for analysis, but how do they do the thresholding? Do they let the software set thresholds or do they adjust them for each sample? Eg You can see from the plots that they are slightly different for the 3 donors.

For donor 20 they say that IPDA-A1D partially captures the SNPs, does this mean that they think they can only quantify part of the intact reservoir? Maybe they can test this with patient-specific degenerate primer/probe sets?

The use of degenerate primer/probe sets results in less good separation between positive and negative partitions: the authors mention that in the part in which they test the probes on JLat cells. Applying this assay to samples where there are even more mismatches than the ones they cover can result in no separation at all. Can the authors further comment on that.

Minor remarks

A type on line 202 for consistency: "near-full length" should be "near-full-length"

Reviewer #3 (Remarks on code availability):

We reviewed the code on intactness, but not on the subtype. That would require more input from my bioinformatic team, hence more time.

Responses to reviewer comments

Reviewer #1 (Remarks to the Author):

The authors carefully and substantially revised manuscript according to the reviewers' comments. They have done adaptation of the IPDA for use in the RHSP cohort in revised paper. Further, the authors carefully validated the efficiency of the IPDA-A1D. They applied the IPDA-A1D to samples derived from participants in the RHSP cohort. IPDA-B failed to detect intact provirus in all cases (n=4), but the IPDA-A1D rescued the detection of intact proviruses. I agree with the authors that optimization of IPDA based on actual sequence data would give us more valuable and practical advantage for the clinical application in near future. As authors mentioned, HIV-1 infection is more evident in Africa, but previous studies so far focused more on HIV-1 subtype B, which is the most common in western countries but not in Africa. HIV-1 provirus characterization in non-B subtypes has not been performed as that in subtype B. Thus, the data presented in this study would be valuable information to think about how HIV-1 infection in Africa would be regulated to reduce HIV burden there. The revised manuscript is much more improved and informative than initial version. I have no additional comments.

We thank reviewer #1 for your positive feedback.

Reviewer #2 (Remarks to the Author):

All of my previous concerns and suggestions were addressed thoroughly. I have no additional questions or concerns. In my view, the manuscript fills an important gap in the field related to expanding the gold-standard approach for characterizing the HIV reservoir in people living with HIV to people are not men with subtype B infection. I commend the authors for including women and people with other subtypes, including recombinant forms, in studies of HIV persistence and cure-related research. The finding that co-infections leading to diverse recombinant forms of HIV is a frequent occurrence in the world is an important contribution to the field.

We thank reviewer #2 for your positive feedback.

Reviewer #3 (Remarks to the Author):

The authors have made significant improvements to the presented manuscript. We commend them for their efforts and sharing more detailed overviews on their methodology and analysis (MOCHI, primer/probe designs and intactness parameters) as requested. Some remarks remain upon revision:

We thank reviewer #3 for your constructive and supportive feedback.

Major remarks

- I would contest the following statement on lines 69-72 and ask to rephrase:

“As such, the suitability of the primer/probe locations and sequences for use in Africa, which is affected almost exclusively by non-B subtypes, remains a major knowledge gap and poses a barrier to cure research in the regions where HIV-1 disease burden is the heaviest globally.”

While proviral reservoir sequences are indeed understudied, the suitability of IPDA primer/probe designs targeting subtypes A and D have been studied before, for example by Cassidy et al. (<https://www.ncbi.nlm.nih.gov/pmc/articles/PMC8786636/>: cross-subtype IPDA) (referenced later in line 260). Please acknowledge other efforts (<https://www.ncbi.nlm.nih.gov/pmc/articles/PMC8786636/>, <https://link.springer.com/article/10.1186/s12985-024-02300-6>)

We appreciate this suggestion. We have rephrased as follows:

“Establishing the suitability of the primer/probe locations and sequences for use in Africa, which is affected almost exclusively by non-B subtypes, will accelerate cure research in the regions where HIV-1 disease burden is the heaviest globally. To date, studies have adapted the IPDA for cross-subtype application inclusive of subtype A, B, C, D, and CRF01_AE (Cassidy *et al.*) and for subtypes B and C (Buchholtz *et al.*). Given that HIV-1 genetic diversity is often region-specific, we characterized proviral genome sequences in people living with HIV in Rakai, Uganda, who were receiving suppressive ART, and leveraged these to design a regionally-adapted IPDA that would maximally capture local HIV sequence variation.” (Line 69-76)

- The statement on lines 126-128 can be misleading. While the authors refer to previous work (using MIP-Seq data with linked IS and provirus) to prove their hypothesis on identical NFL sequences being derived from clonal expansion, this idea doesn't hold when applied to FLIP-Seq data. First of all, the sole proof for a clone is having a linked integration site (which FLIP-Seq doesn't give) and is missing from these datasets. Next, others have shown (<https://doi.org/10.1371/journal.ppat.1005689>) that identical subgenomic sequences (even being NFL) are not direct proof of identical full-length HIV-1 genomes (aka clonal infected cells). While the statement might hold true when applied to a cohort of individuals who initiated therapy late after HIV-1 acquisition, it certainly would not stand when applied to samples from PWH initiating therapy immediately after HIV-1 acquisition due to limited proviral diversity. I suggest to rephrase and clearly mention the limitations of this analysis, and not (misleadingly) state this as a general fact.

To acknowledge that identical FLIP-seq sequences are not conclusive proof of clonal expansion, we have modified all mentions of clonal expansion and clones to “inferred clonal expansion” and “inferred defect/intact clones.” (Results Line 132-137 and Discussion Line 235).

- Concerning the quantification of intactness: the authors report their median total but don't really say anything about the size of the intact reservoir. They do say on line 216 that they were able to obtain intact values in all samples with the new design, but I am curious about the results. The examples in Figure 4 are also quite high, according to the fact that those people are on therapy: donor 7  140 intact/million

donor 20 250 intact/million. it would be a nice addition if the authors reported total and intact for each donor. In this way the reader/reviewer can see for each patient whether it makes sense that they do- or do not- obtain intact sequences with FLIPS.

We thank reviewer #3 for this suggestion. Figure 4 now reports both the total and the intact genomes for each donor in Figure 4. In addition the results now report the range of intact and total values observed in the four donors for whom IPDA-A1D rescued detection both in-text and in Supplementary Table 9:

"Of these four participants, IPDA-A1D yielded between 97 and 254 intact, and between 1477 and 2779 total, HIV copies per million CD4+ T-cells." (Line 220-222)

- In the methods the authors report that they use the quantasoft software for analysis, but how do they do the thresholding? Do they let the software set thresholds or do they adjust them for each sample? Eg You can see from the plots that they are slightly different for the 3 donors.

We visually inspected all raw ddPCR plots and manually adjusted the thresholding as necessary.

Briefly, HIV-negative donor DNA, which represents the double-negative population, was used to guide the location of initial thresholds, which for the IPDA-A1D were normally around $x=2000$ and $y=2000$. These thresholds were then slightly shifted, if necessary, to accommodate the natural variation in fluorescence amplitudes between samples that occurs due to HIV polymorphism, and to maximize discrimination of positive from negative populations. All technical replicates of the same sample used the same threshold. We have added these details to Methods Line 414-420.

- For donor 20 they say that IPDA-A1D partially captures the SNPs, does this mean that they think they can only quantify part of the intact reservoir? can they test this with patient-specific degenerate primer/probe sets?

We appreciate this comment, which also made us realize that our original reporting of Donor 20's variants in the *psi* probe region as adjacent mixed bases ("-----KW") did not allow the reader to understand the number and sequence of variants present, and therefore their potential to be detected by either assay. We have now fixed this by reporting the specific sequence and proportion of each of Donor 20's two within-host variants. We also realized that, in addition to discussing the proportion of Donor 20's reservoir that is detectable by IPDA-A1D, we should also ideally touch upon why IPDA-B fails in this person.

In short, despite Donor 20's within-host variation, our data strongly supports the notion that IPDA-A1D is detecting their full reservoir. But, rather than demonstrate this via the design of custom oligos, we instead drew upon our experience with the IPDA-B - specifically, our knowledge of what polymorphisms are tolerated by this assay, versus which ones cause failure. We adopted this approach as it would also allow us to also confirm why IPDA-B failed in this case.

These new data, and a detailed explanation have been added as Figure S8, reproduced below:

Examination of the resilience of *IPDA-B* probes to the type of within-host HIV diversity seen in Donor 20

Panels A and B show donor 20's co-dominant variant sequences in the *psi* and *env* probe regions, respectively. Dashes indicate a match to both assays, green bases indicate a mismatch to the IPDA-B that is captured by IPDA-A1D, and purple bases indicate mismatches to both assays.

As shown in Figure 4C, Donor 20's proviral pool yielded an *env* failure with IPDA-B, where detection was rescued by IPDA-A1D. Here, we draw from our knowledge of which polymorphisms are (or are not) tolerated by IPDA-B, to deconstruct Donor 20's assay results.

Panels C, D, E, F: The IPDA-B is very tolerant of mismatches near the end of the *psi* probe region. Here, we show representative data from participants whose proviral pools have up to three consecutive "T" mismatches at the end of the *psi* probe region (*panels C, D, E*). All are well-tolerated with minimal impact on amplitude. Though we did not have any examples of a participant with a single G at the penultimate position, participants with a "G" mismatch at the very end are also well tolerated (*panel F*). Reasoning that polymorphisms tolerated by the IPDA-B will also be tolerated by IPDA-A1D, these data support the notion that both IPDA-B and IPDA-A1D detected the *psi* variation in Donor 20.

Panels G, H. The IPDA-B tolerates a single "T" mismatch at the second position of the *env* probe region, though with reduced amplitude (*panel G*). We thus infer that the IPDA-A1D will also tolerate this polymorphism. The IPDA-B however does not tolerate an "A" at the fourth position from the end of the *env* probe; this causes assay failure (*panel H*). As this "A" is found in both of Donor 20's variants, this likely explains why IPDA-B produced an *env* assay failure for this donor (shown in Figure 4C). By contrast, the IPDA-A1D assay design captures this "A" with the degenerate "R" base at this position, which is consistent with our observation that IPDA-A1D rescued detection in Donor 20 (Figure 4C).

Taken together, these observations are consistent with Donor 20's IPDA-B and IPDA-A1D results shown in Figure 4C, and additionally support the notion that IPDA-A1D is likely capable of detecting both of Donor 20's co-dominant proviral species.

- The use of degenerate primer/probe sets results in less good separation between positive and negative partitions: the authors mention that in the part in which they test the probes on JLat cells. Applying this assay to samples where there are even more mismatches than the ones they cover can result in no separation at all. Can the authors further comment on that.

We acknowledge that the reduced separation caused by the degenerate probe design could make gating difficult for samples that have even more mismatches. We now acknowledge this in the discussion as follows:

“The use of degenerate probes in the IPDA-A1D to accommodate HIV diversity reduced the separation between negative and positive ddPCR droplets in both *psi* and *env* channels. It is possible that separation will be even further reduced in samples with additional mismatches, an issue that will need to be monitored as we apply IPDA-A1D to larger populations. IPDA results should always be interpreted in context of HIV's substantial between- and within- host diversity, where the latter can lead to quantification of only a portion of a given individual's reservoir, a scenario that cannot be assessed without individual-level HIV sequencing of every participant. As a result, while IPDA is a robust assay for samples where the viral diversity is constant or known, and fairly robust for longitudinal characterization of samples from individuals on long-term ART, caution should be exercised when applying the IPDA cross-sectionally across individuals or cohorts whose HIV sequences have not been characterized. Wherever possible, alternative

methods such as viral outgrowth assays, FLIP-seq and/or ddPCR that targets different genome regions should be used to support IPDA findings in these cases.” (Line 281-294)

Minor remarks

A type on line 202 for consistency: “near-full length” should be “near-full-length”

Thank you. We have corrected this error.

Reviewer #3 (Remarks on code availability):

We reviewed the code on intactness, but not on the subtype. That would require more input from my bioinformatic team, hence more time.

Please let us know if there are additional concerns regarding MOCHI.

REVIEWERS' COMMENTS

Reviewer #3 (Remarks to the Author):

All concerns have been addressed.